# Emotional Intelligence as a Mediator between Subjective Sleep Quality and Depression during the Confinement Due to COVID-19

**DOI:** 10.3390/ijerph18168837

**Published:** 2021-08-22

**Authors:** María Pilar Salguero-Alcañiz, Ana Merchán-Clavellino, Jose Ramón Alameda-Bailén

**Affiliations:** 1Basic Psychology Area, Department of Clinical and Experimental Psychology, University of Huelva, 21007 Huelva, Spain; pilar.salguero@dpsi.uhu.es; 2Social Psychology Area, Department of Psychology, University of Cádiz, 11519 Puerto Real, Spain; ana.merchan@uca.es; 3INDESS (Research Universitary Institute for Sustainable Social Development), University of Cádiz, 11406 Jerez de la Frontera, Spain

**Keywords:** emotional intelligence, perceived sleep quality, depression, confinement

## Abstract

In March of 2020, as a consequence of the health crisis caused by the SARS-CoV-2 (COVID-19) virus, the State of Alarm and home confinement of the entire population was imposed in Spain. It is foreseeable that this exceptional situation will have psychological effects on citizens. In this work, the impact of confinement on perceived sleep quality and depression is evaluated through questionnaires, as well as the mediating role of Emotional Intelligence (EI) in this relationship. Our results show, firstly, higher prevalence of depressive symptoms in women and young people associated with poorer perceived sleep quality, and secondly, that Emotional Intelligence intervenes as a mediator in this relationship through three different pathways. Worse perceived quality of sleep causes a greater number of depressive symptoms. In addition, this direct relationship may be enhanced by the mediating role of Emotional Intelligence, which we can express in three different ways: low perceived sleep quality and high emotional attention lead to greater depression; low perceived sleep quality and low emotional clarity increase greater symptoms of depression; and low perceived sleep quality together with low clarity and low emotional repair increase levels of depression. Therefore, we can conclude that improving the skills involved in Emotional Intelligence might increase perceived sleep quality, and thus reduce depressive symptoms, which in turn may improve the quality of life.

## 1. Introduction

During the first months of 2020, the SARS-CoV-2 (the virus responsible for COVID-19) was detected in Spain. On March 14, the Spanish Government imposed a State of Alarm, involving a home confinement of the entire population. This unprecedented situation not only posed a medical risk for citizens but also had a psychological, emotional and social impact [1].

From the beginning of the pandemic, different investigations around the world have tried to evaluate the psychological scope of the health crisis in general and the confinement situation in particular. Thus, some of the first studies have shown negative effects on mental health as a result of quarantine for viral infection [2].

Overall, these studies show an increase in anxiety, stress and depression symptoms in confined people, with variations depending on socioeconomic variables, age, sex and other factors such as time elapsed since the beginning of the pandemic [1,3,4,5].

Among the most influential determinants, age and gender have played critical roles. Depressive symptoms ware more frequently observed in women than in men; higher scores in stress, anxiety and depression ware observed in young people (18–25 years), followed by adults (26–60), and scores in these symptoms ware lowest in the population over 60 [1,3,6,7].

Since the end of the last century, the relationship between depression and sleep has been described. In epidemiological studies analyzing the general population, it has been shown that insomnia is twice as prevalent in depressed people as in non-depressed people [8]. When sleep is objectively assessed by polysomnography, a correlation is observed between major depression and increased rapid eye movement (REM) sleep latency [9]. This relationship is confirmed in subjective sleep studies evaluating perceived quality of sleep. These studies reveal that sleep quality is altered in depressed patients [10,11].

It has been observed that both quantity and quality of sleep are related to depressed mood [12]. Specifically, when depressed mood is evaluated, subjects with high sleep quality obtain lower scores than those with medium or low sleep quality. These results are consistent with some studies that show a correlation between sleep quality and psychopathological disorders, especially depression [13,14,15]. This relationship between depressed mood and some parameters of sleep quality and number of sleep hours is also observed in healthy populations [16]. However, some studies show that the quality of sleep, but not the quantity, is related to depressed mood [17].

Therefore, there is an apparent relationship between perceived sleep quality and depression. For this reason, it can be argued that in confined conditions, in which an increase in depressive symptoms is expected, a worsening of the quality of sleep could be observed, and vice versa.

Thus, in a study conducted in China under total confinement conditions, symptoms of post-traumatic stress were observed in 7% of the participants, with higher prevalence in women, and these symptoms were associated with a lower quality of sleep [3]. Following this line, Roy et al. [18], in a study conducted in India, reported that 72% of participants were concerned about COVID-19; 40% reported extreme concern (paranoia) about being infected; and 12% experienced sleep alterations. Likewise, Sandín et al. [7], in a study with Spanish population, observed that the incidence of sleep alteration was more than double in confined conditions for young people (19–30 years, 36.4%), compared to middle-aged groups (31–50 years, 17%) and over 51 years (12.2%).

In this study, we aim to explore the relationship between confinement, sleep (perceived sleep quality) and depression. We also explore how some cognitive processes that act as mediators, especially perceived Emotional Intelligence, can influence the relationship between depression and sleep quality.

Emotional Intelligence (EI) is a concept that links cognition and emotion. It is currently defined as “the ability to perceive, assess and express emotions accurately; the ability to access and/or generate feelings that facilitate thinking; the ability to understand emotions and emotional knowledge; and the ability to regulate emotions promoting emotional and intellectual growth” [19] (p. 5).

EI has been shown to act as a protective factor against depression, for example, in the academic context of university students of different degrees [20,21,22].

However, there is no definitive evidence on the role of EI as a mediating factor between sleep quality and depression during confinement. Some studies exploring the incidence of depression during confinement have shown a higher incidence of depression in confined populations, although with variations mainly depending on sex, age, previous health status and socio-economic conditions [1,5,6].

There is evidence that confinement has an emotional impact on the general population. Since EI may be considered the ability to reason with emotions [19], it seems necessary to find out if the skills to manage emotions can act as mediators against depression and sleep disturbances in confinement situations..

Therefore, the objective of this work is to evaluate how confinement has affected cases of depression in a confined population, considering age and sex as relevant factors. In addition, we explored perceived quality of sleep in this population and the possible role of EI, as emotional information processing, in mediating relevant variables in this relationship.

## 2. Materials and Methods

### 2.1. Participants

In this study participated 188 Spaniards, of which 67.6% were women. Mean age was 46.45 years (SD = 12.37, range = 19–75). The participants were recruited from 7 autonomous communities of Spain: 95.7% from Andalusia, 1.6% from the Community of Madrid, and 0.5% equally distributed from the Valencian Community, Cantabria, Extremadura, Basque Country and Galicia. Most of the sample was married (married = 53.7% and single partner = 5.9%), while 31.4% were single, 8.5% were divorced, and 0.5% were widowed. Finally, 49.5% of the sample had studied at university, 35.1% had completed secondary studies or professional training, 5.9% had completed postgraduate studies, 4.8% had finished primary studies, 4.3% had completed doctoral studies, and only 0.5% of the sample had not completed any studies.

### 2.2. Instruments

The questionnaire used included the scales that are explained below, in addition to specific questions about age, sex, marital status, the autonomous community in which they were living and level of studies.

Subjective sleep quality was assessed using a version of the Pittsburgh Sleep Quality Index (PSQI) [11] translated and validated by Royuela and Macías (1997) [23]. In this questionnaire, scores are calculated using a Likert scale ranging from 0 to 3, in which 0 means very good quality while 3 means very poor subjective sleep quality. The PSQI has an alpha value of 0.80.

The Trait Meta-Mood Scale (TMMS-24) [24] was used for the evaluation of Emotional Intelligence. This questionnaire assesses the perception or beliefs about one’s emotional abilities. It contains 24 items, based on a 1–5 Likert scale. TMMS-24 is divided into three dimensions, each consisting of 8 elements: emotional attention (ability to identify one’s emotions and emotions of others, and the ability to know how to express them), emotional clarity (understanding of emotions) and emotional repair or regulation (ability to handle emotions). The reliability and validity indices reported are adequate; the alpha value is 0.90 for the attention dimension, 0.90 for clarity and 0.86 for emotional repair [25]. In our sample, the alpha value was 0.913 for the attention dimension, 0.942 for clarity and 0.922 for emotional repair.

Depression was measured with the Beck Depression Inventory-II (BDI-II) [26]. It is a self-reported measure, with a total of 21 items that explore the severity of depressive symptoms. In each item, there are 4 response alternatives, regarding perceived emotional state in the last two weeks. Each item is scored from 0 to 3; therefore, the total score ranges from 0 to 63 points (scores lower than 13 points suggest minimal depression; between 14 and 19 points, mild depression; between 20 and 28 points, moderate depression; and scores higher than 29 suggest severe depression). Considering the factorial analysis conducted by Sanz and García-Vera (2013) [27], aimed at evaluating if this inventory has unifactorial or bifactorial properties, and assuming that with the total score any information for the two factors should be included, unifactorial analyses were conducted. The total scale has shown adequate reliability, with a Cronbach’s alpha value of 0.913.

### 2.3. Procedure

Data from the questionnaires were collected online via a questionnaire built in Google Docs. The data collection period took place during the State of Alarm imposed by the Spanish Government from 14 March to 21 June 2020. We mainly used the Moodle virtual education platforms of the Universities of Cádiz and Huelva teaching during this period was conducted online through these platforms). In addition, the questionnaire was communicated through social networks (Facebook and Twitter) and participants were asked to disseminate it.

There was only a single inclusion criterion: all participants had to be of legal age. Throughout the procedure, they were informed about the objectives of the investigation and any other issues regarding the study. In the same data collection questionnaire, before starting the data collection, the participants were informed about the objective of the study, i.e., the influence of confinement on aspects such as sleep quality and its relationships with variables such as Emotional Intelligence. They were also informed that participation was completely voluntary and that they could stop completing the questionnaire at any time, without any explanation, and that no information would be collected in that case. Upon completing the questionnaire, they were informed that all the data obtained would be treated anonymously, confidentially and exclusively for the specific purposes of this study.

Before starting to record their responses, all participants provided an informed consent before participation in this study, which was voluntary and confidential. The study was carried out according to the last revision of the Declaration of Helsinki.

### 2.4. Analysis of Data

For data analysis, the statistical program SPSS 25 and the PROCESS macro were used. First, descriptive statistics (means and standard deviations) were analyzed. To analyze sex differences, non-parametric tests were conducted for two independent samples. Spearman’s Rho correlations and the reliability index of the scales were also calculated by Cronbach’s alpha coefficient.

Via the PROCESS macro, mediation analysis was established with a 95% confidence interval and several bootstrapping samples of 10,000. The estimates of each analysis were made through their respective non-standardized regression coefficients (Coeff), standard errors (SE), t-values and significance levels (p), as well as the different values of the lower (LLCI) and upper (ULCI) limit of the confidence interval. The interpretation of significance was based on the lower and upper limit values of the confidence interval. Thus, 0 values confirmed no significance. All analyses were carried out using the SPSS package (version 25; IBM, Chicago, IL, USA).

We used PROCESS Model 6 to examine the direct and indirect effects of subjective sleep quality on depression. The mediation analyses were conducted using the Emotional Intelligence dimensions as mediators of this relationship. We analyzed whether the effect of the independent variable (X) (subjective sleep quality) on the dependent variable (Y) (depression) could be mediated by the mediating variables (M) (the three dimensions of the EI). As shown in Figure 1, the parameter (c’) is the direct effect of X on Y, controlling the mediating variable, the parameters (a 1,2,3) represent the direct effect of X on each M, the parameters (b 1,2,3) represent the direct effect of each M on Y. The total indirect effect (d) refers to the relationship between the three mediators (d21, d32 and d31); the specific indirect effect (a1b1, a2b2 and/or a3b3) refers to the role of a specific mediator in the relationship between subjective sleep quality and depression; and the total effect (c) is the sum of the direct and indirect effects, that is, when the mediators are excluded from the regression. Sex and age were included in the model as covariates.

## 3. Results

### 3.1. Preliminary Analyses

Descriptive statistics of some variables are here reported, both for the total sample and separately for men and women (Table 1). Statistically significant differences were observed in subjective sleep quality (Z = −3.85; *p* = 0.000), emotional attention (Z = −2.02; *p* = 0.043) and levels of depression (Z = −3.05; *p* = 0.002). As depicted in Table 1, women obtain, in general, higher scores in all significant parameters compared to men. These include higher levels of depression, worse levels of subjective sleep and higher levels of perceived emotional attention. However, there are no significant sex differences with respect to clarity or emotional repair (*p* > 0.05).

Table 1 also shows the correlations between all the main variables. The results indicate some relationships. Specifically, subjective sleep quality is positively correlated with emotional attention and depression, and is negatively correlated with age (*ρ* = −0.186, *p* = 0.011) and emotional clarity.

The BDI II total score shows a mean score of 6.51 in men and 10.10 in women, both within the minimal depression range. The highest individual score was 38 points. In terms of severity of depression, 73% of the participants showed minimal depression, 15.3% showed mild depression, 8.5% moderate depression and the remaining 3.2% severe depression. The BDI II total score also reveals positive correlations with emotional attention and negative correlations with emotional clarity, emotional repair and age (*ρ*= −0.214; *p* = 0.003). Finally, regarding the interactions between EI variables, there were positive correlations between attention and clarity, and between clarity and emotional repair.

### 3.2. Mediation Analysis

The analysis of the EI serial mediation model (Attention, Clarity and Repair) to explore the relationship between subjective sleep quality and depression is shown in Table 2. In the first regression analysis, subjective sleep quality represented 25.9% of the variance in depression levels (*R^s-q^* = 0.259, *F* = 21.45, *p* < 0.01), considering age as a covariate, *b* = −0.104, 95% CI [−0.182; −0.026]. However, 36.6% of the total variance in depression was explained by the global model, which included subjective sleep quality and the three mediators of Emotional Intelligence (*R^s-q^* = 0.366, *F* = 17.44, *p* < 0.01), considering age as a covariate, *b* = −0.104, 95% CI [−0.182; −0.026].

The results presented in Table 2 show that the total effect (c) and the direct effect (c‘) of subjective sleep quality on depression are significant. According to the regression coefficient, which is based on the fact that the 95% CI of the point estimate does not include the value 0, the results show the mediation of indirect effects, obtaining three specific indirect effects through *Ind1*, *Ind2* and *Ind6*. Regarding the significant indirect effects when comparing between pairs, there were no significant differences because the values of each lower and upper limit of the confidence intervals contain the value 0. Therefore, it can be determined that the three mediation pathways have the same level of importance in the model (see Figure 2). The first pathway (*Ind1*) indicates that a poorer subjective sleep quality along with high levels of attention will lead to higher levels of depression. Pathways two and three (*Ind2* and *Ind6*) reveal that poorer subjective sleep quality along with lower levels of emotional clarity lead to higher depression, with model three showing poorer clarity and worse emotional recovery.

## 4. Discussion

The objective of this work was to determine the relationship between perceived sleep quality and depressive symptoms, as well as the mediating role of EI in this relationship, and this in the extraordinary circumstances of home confinement due to the COVID-19 health crisis.

Our results regarding levels of depression in confinement conditions are in line with previous reports, particularly in two senses; firstly, women have shown higher levels of depression than men and, secondly, during confinement, higher levels of depression have been observed in young populations [1,3,6,7].

On the other hand, the close relationship between perceived sleep quality and depression, described in the literature both in populations with psychopathology [13,14,15] and in healthy populations [16,17], is kept under confinement conditions, as expected.

According to our results, EI modulates the relationship between perceived sleep quality and depression, as revealed by the correlation and mediation analyses.

Regarding the correlation analyses, a negative correlation between subjective sleep quality and depression was revealed, in which a positive correlation between emotional attention and depression was observed. Particularly, a higher number of depressive symptoms is associated with poorer sleep quality and better emotional attention. A negative correlation between depression and emotional clarity was found. The correlation between sleep quality and depression was expected during the confinement period and is consistent with previous studies under normal conditions [13,14,15,16,17].

The correlations between depression and the different EI components are also in line with previous results showing that emotional attention correlates positively with different dimensions of sadness (cognitive, physiological and suicidal tendency), while also showing that clarity and emotional repair correlate negatively with depressive symptoms [28]. Extremera et al. (2006) [29], using the Beck Inventory, have also found that depression correlates positively with emotional attention and negatively with clarity and repair.

This suggests that attention to emotions is beneficial for the individual provided it includes an adequate management of the emotional information, through clarity and repair. However, a high level of attention along with psychological discomfort can lead to more depressive symptoms. Therefore, a high score in emotional attention could be a necessary, but not sufficient, condition for adequate emotional self-management.

The results of the present study show that depression is negatively correlated with age under confinement conditions, and this finding is not congruent with the relationship found between these variables under normal conditions. Prevalence studies show that the incidence of depression in people under 25 years old is half that in the 25–45 year-old age group, and lower than in the 45–55 year-old age group (14.1%, 33.1% and 25%, respectively) [30]. Our results under confinement conditions are consistent with previous studies and confirm that depression is more prevalent in young people [1,3,6,7].

Our mediation analyses show that EI is a relevant mediating factor that modulates the relationship between perceived sleep quality and depression under confinement conditions. This mediation can be established in different ways:-Firstly, poor sleep quality plus high emotional attention lead to more depressive symptoms. This effect was expected since both poor sleep quality and high emotional attention lead to higher levels of depression. Thus, both factors together have the potential to induce similar effects.-Secondly, poor subjective sleep quality, along with low levels of emotional clarity, lead to higher levels of depression. The explanation could be similar to the previous case, that is, poor sleep quality is related to more depressive symptoms, and low levels of emotional clarity are also related to more depressive symptoms, indicating that both factors together may also influence levels of depression.-Thirdly, poor subjective sleep quality, along with low clarity and emotional recovery, implies higher levels of depression. This relationship is also predictable since low levels of clarity and emotional repair, as well as poor perceived sleep quality, are closely related to a high number of depressive symptoms.

Some limitations of this work can be raised. Firstly, the data have been collected through self-reports, which can be a significant source of bias. On the other hand, data collection was carried out in a specific period of confinement and not during the entire confinement. Therefore, there could be differences in terms of perceived sleep quality and depressive symptoms between the first and last weeks of confinement. These circumstances were not possible in this work since the data collection was carried out in the second half of the confinement period. Possibly, multiple other variables may affect sleep quality during the period of confinement, such as overexposure to television and electronic devices, alterations in schedules and eating habits, as well as the intake of drugs and alcohol. In addition, the social isolation inherent in confinement may also contribute to higher levels of depression.

## 5. Conclusions

This study shows that during the confinement period in Spain due to the pandemic, higher levels of depression are observed in women and younger people. This depressive symptomatology is accompanied by a worse perceived sleep quality. This relationship seems to be mediated by the different dimensions of Emotional Intelligence.

Interventions aimed at improving sleep quality, through skills related to EI, could be effective in reducing depressive symptoms and thus help to substantially improve people’s quality of life. This intervention should involve three EI dimensions, that is, it would be necessary to decrease emotional attention and additionally increase both clarity and emotional repair.

Finally, future research should evaluate possible long-term psychological effects induced by both the period of home confinement and the restrictions upon social life and work conditions as a result of the COVID-19 pandemic.

## Figures and Tables

**Figure 1 ijerph-18-08837-f001:**
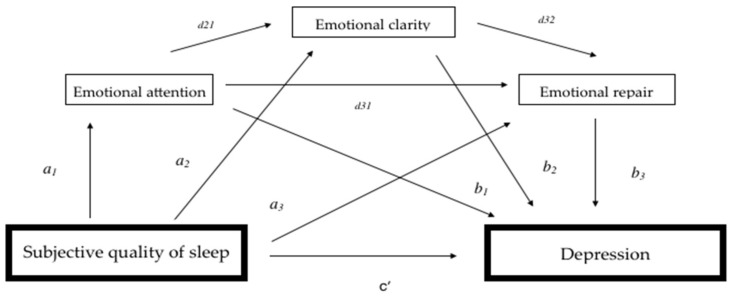
Conceptual and statistical diagram of the serial mediation of Emotional Intelligence in subjective sleep quality and depression.

**Figure 2 ijerph-18-08837-f002:**
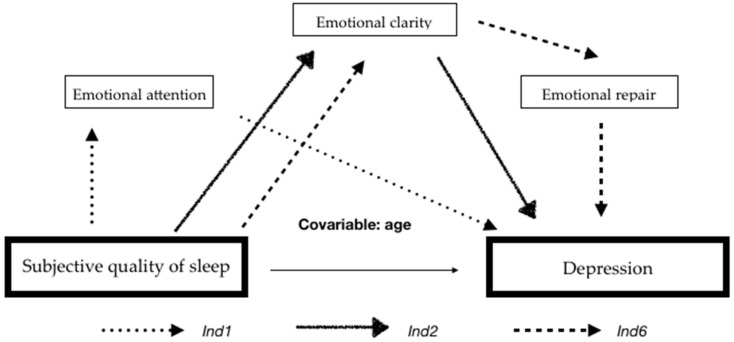
Mediation model in the relationship between subjective sleep quality and depression.

**Table 1 ijerph-18-08837-t001:** Means, standard deviations and bivariate correlations between the study variables.

Instruments	Men		Women		Overall		Spearman’s Rho Correlations			
	*M*	*SD*	*M*	*SD*	*M*	*SD*	1	2	3	4
1. Subjective sleep quality	1.11	0.61	1.57	0.81	1.43	0.78	-			
2. Emotional Attention	21.25	7.12	23.63	7.30	22.86	7.31	0.194 **	-		
3. Emotional Clarity	25.51	7.46	25.20	7.86	25.30	7.72	−0.143 *	0.450 **	-	
4. Emotional Repair	26.21	7.67	25.78	7.25	25.92	7.37	−0.048	0.333 **	0.618 **	-
5. Depression	6.51	6.91	10.10	8.51	8.95	8.19	0.528 **	0.200 **	−0.198 **	−0.176 *

* = *p* < 0.05; ** = *p* < 0.01.

**Table 2 ijerph-18-08837-t002:** Results of the analysis evaluating serial mediation of Emotional Intelligence in subjective sleep quality and depression, including sex and age as covariates.

Path	Coefficient	SE	Boot LLCI	Boot ULCI	*t*	*p*
Total effect (*c*)	4.448	0.701	3.065	5.832	6.346	0.000
Direct effect (*c*′)	3.377	0.689	2.018	4.737	4.901	0.000
*a_1_*	1.473	0.707	0.0775	2.868	2.082	0.039
*a_2_*	−2.625	0.665	−3.938	−1.312	−3.945	0.000
*a_3_*	0.190	0.593	−0.979	1.360	0.321	0.748
*b_1_*	0.353	0.078	0.199	0.508	4.521	0.000
*b_2_*	−0.218	0.089	−0.393	−0.043	−2.462	0.015
*b_3_*	−0.186	0.086	−0.356	−0.017	−2.166	0.032
*d* _21_	0.514	0.068	0.379	0.649	7.499	0.000
*d* _31_	0.078	0.067	−0.054	0.211	1.169	0.244
*d* _32_	0.574	0.063	0.449	0.699	9.083	0.000
Indirect effects	**Effect**	**SE**	**Boot LLCI**	**Boot ULCI**		
Total indirect effect	1.071	0.375	0.401	1.861		
*Ind1: a_1_b_1_*	0.521	0.292	0.015	1.159		
*Ind2: a_2_b_2_*	0.573	0.259	0.126	1.141		
*Ind6: a_2_d_32_b_3_*	0.281	0.146	0.052	0.619		

**Notes: Abbreviations:** Boot LLCI: bootstrapping lower limit confidence interval; Boot ULCI: bootstrapping upper limit confidence interval; SE: standard error.

## Data Availability

The dataset is available in the repository.

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
