# Peer review of "Emotional Intelligence as a Mediator between Subjective Sleep Quality and Depression during the Confinement Due to COVID-19"

_ijerph, 2021, doi:10.3390/ijerph18168837_

Round 1
Reviewer 1 Report
This manuscript presents the study on human participants from Spain on the sleep quality during COVID-19 restriction. The study correlates the sleep quality with emotional intelligence. Standard self-reported questionnaires were used in this study.
- The study presents the hot topic of 2020 COVID-19 pandemic. However, I feel the findings are not new, and has been reported by several researchers in the different region.
- In general, manuscript requires correction in language. It should provide more methodology details on the collection, analysis and interpretation of data.
- How were the participants approached during the pandemic restrictions? Please specify
- Please clearly specify the online data collection method.
- “Throughout the procedure, they were informed about the objectives of the investigation and any other issue regarding the study.” Please specify the mode of information transformation to participants.
- Check spelling mistakes and spacing errors throughout the manuscript. Please refer to attached file with highlights in yellow color.
- Please remove limitations from conclusion. I suggest presenting limitation separately before conclusion or in the later part of discussion.
- Please add all major limitation of these study. TV or electronics usage during pandemic may also affect the sleep, and change engagement. This could potentially interfere with the conclusion of this study. Also, changes in the eating schedules and patterns of eating during restriction can potentially affect the sleep and measurement in this study. Likewise, ongoing medication or changes in medication intake can potentially affect sleep and the study measurements.
- Table 2: what is “Model : 6; Y : Depression; X : Subjective sleep quality; M1 : Emotional attention; M2 : Emotional clarity; M3 : Emotional repair;” ? I could not find these in table.

Author Response
Thank you for your comments. I would like to start by apologizing for this delay, as I mentioned to the editor, one of us has had a health problem that has required hospitalization and "absolute" rest, which has delayed us in making the corrections.
We will now respond to all your considerations:
1.- The study presents the hot topic of 2020 COVID-19 pandemic. However, I feel the findings are not new, and has been reported by several researchers in the different region.
-Indeed, we address a highly analyzed topic in relation to the COVID-19 pandemic and in fact the problems related to sleep during confinement have been a very prominent element, which has aroused much interest in their research in different territories. Our contribution has been to use mediation analysis with emotional intelligence in its relationship with sleep.
2.- In general, manuscript requires correction in language. It should provide more methodology details on the collection, analysis and interpretation of data.
-The manuscript has been translated and revised by a bilingual professional, however I imagine that it can be improved and we will try to do so.
-We have further clarified the methodological aspects, the analysis and the interpretation of data. Indeed, in some cases they were written in a concise and uninformative way, we hope they are more understandable. We will indicate the list of the changes made in the following points.
3.- How were the participants approached during the pandemic restrictions? Please specify.
Thanks for the observation, it was indeed unclear, we have added in the "procedure" section: "We mainly use the moodle virtual education platforms of the universities of Cádiz and Huelva (it must be remembered that teaching in that period was online through these platforms). In addition, the questionnaire was communicated through social networks (facebook and twitter), while the participants were asked to disseminate."
4.- Please clearly specify the online data collection method.
-Indeed this data was not reported, now we have added in the procedure section that we have used a questionnaire built in google docs.
5.- Throughout the procedure, they were informed about the objectives of the investigation and any other issue regarding the study.” Please specify the mode of information transformation to participants.
Thank you for your comment, we have expanded the information by adding in the procedure section: In the same data collection questionnaire and before starting the data collection, the participants were informed about our interest in analyzing the incidence of confinement in aspects such as sleep quality and its relationships with variables such as emotional intelligence. They were also informed that participation was completely voluntary and that they could withdraw from the questionnaire at any time, without giving explanations and that no information would be collected from it. Upon completing the questionnaire, they were informed that all the data obtained would be treated anonymously and confidentially and exclusively for the specific purposes of this study.
6.- Check spelling mistakes and spacing errors throughout the manuscript. Please refer to attached file with highlights in yellow color.
We have corrected all mistakes marked in yellow and some more. Thanks for your work, this is a problem that sometimes arises when transferring documents from windows to mac and vice versa, it also appears sometimes when transferring the document to pdf.
7.- Please remove limitations from conclusion. I suggest presenting limitation separately before conclusion or in the later part of discussion.
Thanks for the suggestion, we have changed the limitations at the end of the discussion.
8.- Please add all major limitation of these study. TV or electronics usage during pandemic may also affect the sleep, and change engagement. This could potentially interfere with the conclusion of this study. Also, changes in the eating schedules and patterns of eating during restriction can potentially affect the sleep and measurement in this study. Likewise, ongoing medication or changes in medication intake can potentially affect sleep and the study measurements.
-Indeed there are more limitations than those outlined, we have incorporated the following: Point out that there are multiple variables that can affect sleep during the confinement period, among them the excessive use of television and electronic devices, schedules and eating habits, as well as the intake of drugs and alcohol. In addition, the social isolation suffered during confinement can also contribute to causing higher levels of depression.
9.- Table 2: what is “Model: 6; Y : Depression; X : Subjective sleep quality; M1 : Emotional attention; M2 : Emotional clarity; M3 : Emotional repair;” ? I could not find these in table.
-Thanks for detecting this mistake, that information should not appear, it has been removed.
Reviewer 2 Report
This manuscript, titled “Emotional Intelligence as a Mediator between Subjective Sleep Quality and Depression during the Confinement due to COVID-19,” analyzed the association between sleep quality and depression and how emotional intelligence mediates the association.
The study examined an important research question. However, there are several major issues that need to be addressed.
- Emotional intelligence was measured as a trait. It should be noted that traits should not be considered mediators. Mediators should be variables that have malleable characteristics.
- The rationale of including emotional intelligence as mediators in the study should be explained in Section 1. Also, Section 1 should explain the sequence of the three mediators. Need to keep in mind that predictors should temporally precede the first mediator, which temporally precedes the second mediator, and so on.
- There are other variables that could affect depression, for instance, social isolation. The discussion section may discuss the implications of not including such variables in the study.
- The discussion section should be enriched with more in-depth discussion.
Other minor issues include:
- The psychometric properties of the scales should be reported.
- The wording in the abstract (lines 18-21) is rather misleading. It seems to suggest moderation effects instead of mediation effects.
- There are many instances where spacing is missing between two words, e.g., anincrease (line 37), populationover (line 43), somestudies (line 55).
- Line 218 should be Section 4. And hence line 273 should be Section 5.
Author Response
Thanks for your comments. First I would like to apologize for this delay, as I mentioned to the editor, one of us has had a health problem that required a few days of hospitalization, she is already at home although she has to rest "absolute", this has inevitably delayed us .
Here is a response to his comments:
- Emotional intelligence was measured as a trait. It should be noted that traits should not be considered mediators. Mediators should be variables that have malleable characteristics.
We evaluate EI through TMMS-24, which is a self-perception questionnaire of skills involved in EI, and if skills are modifiable, we do not measure EI as a trait. We do not consider IE a trait, but rather an ability to manage emotions, we are based on the concept of IE by Mayer and Salovey.
- The rationale of including emotional intelligence as mediators in the study should be explained in Section 1. Also, Section 1 should explain the sequence of the three mediators. Need to keep in mind that predictors should temporally precede the first mediator, which temporally precedes the second mediator, and so on.
We consider that the confinement has had an emotional impact on the population. Based on this emotional impact, we start from the concept of IE Mayer and Salobey, of considering that EI is the ability to reason with emotions, so we consider it necessary to check if EI mediates between sleep quality and depression.
At the end of the introduction we have incorporated the following:
Confinement had an emotional impact on the population. EI is considered to be the ability to reason with emotions (19), hence the need to find out if the skills to manage emotions act as mediators against depression and sleep disturbances. Based on the original theory of IE (19), the sequence of the three mediators would be Attention, Clarity and Replacement.
- There are other variables that could affect depression, for instance, social isolation. The discussion section may discuss the implications of not including such variables in the study.
Indeed there are a series of variables that we have not analyzed and that may be relevant associated with depression. For this reason we have incorporated at the end of the discussion a reflection on the limitations of our study, in it we comment "Some limitations of this work can be raised. First, the data has been collected through self-reports, which can be a significant source of bias. On the other hand, data collection was carried out in a specific period of confinement and not during the entire confinement. Therefore, there could be differences in terms of perceived sleep quality and depressive symptoms between the first and last week of confinement. These circumstances were not possible in this work since the data collection was carried out in the second half of the confinement period. Point out that there are multiple other variables that can affect sleep during the period of confinement, among them the excessive use of television and electronic devices, schedules and eating habits, as well as the intake of drugs and alcohol. In addition, the social isolation suffered during confinement can als o contribute to causing higher levels of depression.
- The discussion section should be enriched with more in-depth discussion.
We agree that the discussion should clarify the role that EI plays, for this reason we have added the following reflection (l.254-260)
"In this paper we have evaluated if EI modulates the relationship between perceived sleep quality and depressive symptoms. For it, we consider EI as a set of self-perceived skills about how the person manages their own emotions, which means that EI is modifiable, namely, EI is susceptible to improvement through psychological interventions.
In this sense, our results indicate that EI intervenes by modulating the relationship between perceived sleep quality and depression, as revealed by the correlation and mediation analyzes. "
what is connected with what was stated in the introduction and the conclusions, how by working with EI we can influence the quality of life, in this case in the perceived quality of sleep.
Other minor issues include:
- The psychometric properties of the scales should be reported.
We have incorporated the corresponding alphas from the different questionnaires, The PSQI have an alpha value of 0.80 (l. 118-119).
The Trait Meta ‐ Mood Scale (TMMS ‐ 24) [24] The reliability and validity indexes reported are adequate, the alpha value is .90 for the attention dimension, .90 for clarity, and .86 for emotional repair [25] (l .126-127) and Beck Depression Inventory-II (BDI-II) [26]. The total scale has shown adequate reliability, with a Cronbach's alpha value of .913 (0.912 in our sample) (l.138)
- The wording in the abstract (lines 18-21) is rather misleading. It seems to suggest moderation effects instead of mediation effects.
Gracias por el comentario, realmente si resulta complicado de leer. Hemos simplificado la redacción intentando dejarlo más claro, antes decíamos "...low perceived sleep quality accompanied by low emotional clarity causes greater symptoms and depression and third, low perceived sleep quality together with low clarity and emotional repair leads to higher levels of depression that the relationship between perceived."
y lo hemos cambiado por "...low quality of sleep and depression is increased by the mediating role of EI, in three different ways: high emotional attention, low emotional clarity and low emotional clarity and emotional repair."
- There are many instances where spacing is missing between two words, e.g., anincrease (line 37), populationover (line 43), somestudies (line 55).
Thanks for warning. It usually happens to us when we pass a document from windows to mac and vice versa, and even when we pass a document to pdf. We have corrected all these mistakes.
- Line 218 should be Section 4. And hence line 273 should be Section 5.
-Indeed the discussion and conclusions sections were wrongly numbered, this error has been corrected by properly numbering "4. Discussion" and "5. Conclusions"
Reviewer 3 Report
Congratulations, it is w very interesting paper.
I can see that the Authors referred to the first review of another Reviewer.
This paper is very good for me. I just have a few suggestions:
- Emotional Intelligence is written with large letter in the text and a small letter in the title, please standardize it.
2.Line 15 please add (EI) after “Emotional Intelligence”.
- According to BDI II, what is the cut-off point? In results Authors present mean value of BDI II. How do you diagnose e.g mild or severe depression according to total score od BDI II? How many people had no depressive symptoms at all?
- There are no gaps in the references before the dates in each publication
Thank you.
Author Response
Thank you for your motivating comments. We respond to your suggestions below.
1.- Emotional Intelligence is written with large letter in the text and a small letter in the title, please standardize it.
Indeed the term Emotional Intelligence, was treated inconsistently throughout the text, we have chosen to always use "Emotional Intelligence"
2.- Line 15 please add (EI) after “Emotional Intelligence”.
Thank you for detecting this error, it has been corrected and it has been incorporated (EI) in line 15
3.- According to BDI II, what is the cut-off point? In results Authors present mean value of BDI II. How do you diagnose e.g mild or severe depression according to total score od BDI II? How many people had no depressive symptoms at all?
Indeed the information was incomplete. In the description of the BDÇI-II we have incorporated the cut-off points, incorporating the following text (Ln 130-132):
... (13 points, means minimal depression; between 14 and 19, mild depression; between 20 and 28, moderate depression; and from 29, severe depression.).
In the results section we have incorporated the following (Ln 204-208):
The BDI II total score show an average of 6.51 in men and 10.10 in women, in both cases we would be in minimal depression. Analyzing the scores, the highest value obtained is 38 points. According to the severity of the depression, we have that 73% of the participants manifest minimal depression, 15.3% present mild depression, 8.5% moderate depression and the remaining 3.2% severe depression.
4.- There are no gaps in the references before the dates in each publication
Thank you for informing us of this mistake, it has been corrected and the missing space before the year has been incorporated in all references
Round 2
Reviewer 1 Report
The manuscript has been improved and most of the comment has been answered.
However, the spacing error remains in the manuscripts.
Author Response
The manuscript has been improved and most of the comment has been answered.
However, the spacing error remains in the manuscripts.
Thanks for your comments, apologize for the spacing errors reappear, we imagine that it will be a problem when going from word to pdf. We have thoroughly reviewed the word document and we have marked in red all the errors that you kindly report to us in the first round, we hope that now they appear as corrected.
Reviewer 2 Report
Some comments are still not adequately addressed in the manuscript. In the separate “Authors’ response” document, the authors have addressed the comments well (albeit not convincing). Those responses should appear in the manuscript.
The comments that still need clarification in the manuscript include:
- Emotional intelligence was measured as a trait. The authors should specify the malleability of the construct measured.
- The rationale of including emotional intelligence as mediators in the study should be explained in Section 1. Also, Section 1 should explain the sequence of the three mediators. Need to keep in mind that predictors should temporally precede the first mediator, which temporally precedes the second mediator, and so on.
- There are other variables that could affect depression, for instance, social isolation. The discussion section may discuss in detail the implications of not including such variables in the study.
- The discussion section should be enriched with more in-depth discussion.
Other minor issues include:
- The wording in the abstract (lines 18-21) still needs more clarity.
- Line 235 should be Section 4. And hence line 301 should be Section 5.
Author Response
Some comments are still not adequately addressed in the manuscript. In the separate “Authors’ response” document, the authors have addressed the comments well (albeit not convincing). Those responses should appear in the manuscript.
The comments that still need clarification in the manuscript include:
- Emotional intelligence was measured as a trait. The authors should specify the malleability of the construct measured.
Following your suggestions we have developed more in section 1 on the EI model as well as the dimensions of the TMMS, trying to better clarify the naturalization of the measurements.
We have incorporated the following:
We start from the framework proposed by Mayer and Salovey [19] that has allowed an approach to the study, understanding and emotional management, and provoking interest in establishing the predictive capacity and influence of EI in various vital areas. This model considers EI an example of ability, that is, it focuses exclusively on the emotional processing of information and the study of capacities related to it. Conceives EI as an adaptive process of the use of our emotions in cognition to be able to solve problems and adapt effectively to a changing environment.
From this model, the Trait Meta Mood Scale (TMMS) is proposed as a measure of self-perceived EI, which implies that its measurements are highly malleable. As you interact with the environment with new emotionally intelligent behaviors, the way is built to turn them into habits. Thus, these new habits will gradually modify or build new behaviors. EI in this model is ultimately a powerful adaptive tool.- The rationale of including emotional intelligence as mediators in the study should be explained in Section 1. Also, Section 1 should explain the sequence of the three mediators. Need to keep in mind that predictors should temporally precede the first mediator, which temporally precedes the second mediator, and so on.
Indeed, this aspect was presented in a very concise way, we have tried to explain it more clearly, now it appears as follows:
We are interested in exploring how emotional skills allow, first, to observe, identify and think about emotions and feelings, assess and examine their affective states and focus and maximize their emotional experience (emotional attention). Second, identify and describe the emotions that are experienced on a daily basis (emotional clarity), that is, the ability to name our emotions, beyond knowing that we feel good or bad. And third, interrupt negative emotional states and / or prolong positive ones (emotional repair). It is a process of emotional regulation, evaluating attempts to reverse negative emotions in a more positive direction. These three processes can be considered serial or hierarchical, so any emotional intervention begins with emotional attention, a correct emotional identification (emotional clarity) and ends if it is necessary with emotional repair.
We have also tried to make it clearer in the procedure, before we put "...could be mediated by the mediating variables (M) (the three dimensions of the EI)."
We have changed it to "...could be mediated by the mediating variables (M) (the three dimensions of the EI, in this order: emotional attention, emotional clarity and emotional repair). "
- There are other variables that could affect depression, for instance, social isolation. The discussion section may discuss in detail the implications of not including such variables in the study.
Indeed, social isolation during confinement has been a recurring complaint, in addition this isolation affects both the quality of sleep and depression, however in our data there is a low percentage of people who have passed the confinement alone (although this does not mean say that they have not undergone some degree of isolation).
To try to clarify it we have added:
Furthermore, social isolation and the limitation of social interactions caused by confinement can be a very relevant variable. The social isolation caused by different pandemics (SARS-CoV, MERS-CoV, influenza A / H1N1, among others), has had a negative effect on mental health, causing a higher prevalence of different mental symptoms such as anxiety, sadness, low self-esteem, irascibility, emotional lability, among others, and in relation to our theme, it is related to insomnia and depression. In our data, the percentage of people who have passed confinement alone is very low (17.49%), however, we cannot affirm that these people were socially isolated, e.g. they were able to maintain job activity. In addition, making this variable operational during confinement is a bit complicated, since virtual contacts have intensified during the pandemic, although obviously is not the same. Furthermore, social isolation can be both the cause and the effect of depressive symptoms, hence the importance of being cautious with this variable. That is, we cannot rule out that depressive symptoms are the cause of possible social isolation (and vice versa) even when one person lives with others.
- The discussion section should be enriched with more in-depth discussion.
In response to this suggestion and also related to the previous ones we have tried to improve the discussion, thus,
We have tried to analyze in more detail the results of the correlations (ln 288-302) incorporating the following paragraph:
"Regarding correlation analyzes, a positive correlation was revealed between sub-jective sleep quality and depression. This indicates that the low quality of sleep, the depressive symptoms increase, which is expected during the period of confinement, and is consistent with previous studies under normal conditions [13-17]. The three dimensions of TMMS correlate unevenly with sleep quality and depression. Thus, we observe a positive correlation of these variables with emotional attention and a negative correlation with emotional clarity, while emotional repair negatively correlates with depression. In this sense, according to the TMMS, that people with moderate levels of emotional attention carry out more adaptive emotional regulation strategies than those with high levels. The negative correlation of emotional clarity with sleep quality and depression makes sense since high scores in emotional clarity are associated with different health dimensions, such as better self-esteem and interp ersonal satisfaction, less vulnerability to stress and less depression. Finally, emotional repair reflects response-focused emotion regulation since this ability is aimed at modulating the emotional reaction once the emotion has been generated, which may explain the negative correlation with depressive symptoms [28,29]. "
In the case of the mediation analysis, we have detailed the explanations a little more, thus, in relation to emotional attention we have added (ln. 334-335) "As we discussed earlier, it is precisely the moderate levels of emotional attention that carry out more adaptive emotional regulation strategies. ". Related to emotional clarity, we have added (ln. 340-341) "They are precisely high scores in emotional clarity that are positively associated with different dimensions of health." and in relation to emotional repair we have incorporated (ln. 345-347) "There are several works that show how emotional repair has shown a predictive value together with emotional clarity in different areas [20, 24, 25, 29]."
Other minor issues include:
- The wording in the abstract (lines 18-21) still needs more clarity.
We have changed the wording looking for more clarity in what was expressed
before it was written:
"Low quality of sleep and depression is increased by the mediating role of Emotional Intelligence, in three different ways: high emotional attention, low emotional clarity and low emotional clarity and emotional repair."
Now the wording is as follows
"A worse perceived quality of sleep causes greater depressive symptoms, in addition, this direct relationship is increased by the mediating role of Emotional Intelligence, which we can express in three different ways: low perceived sleep quality and high emotional attention leads to greater depression; low perceived sleep quality and low emotional clarity, is associated with greater symptoms of depression, and; low perceived sleep quality together with low clarity and low emotional repair is accompanied by higher levels of depression."
- Line 235 should be Section 4. And hence line 301 should be Section 5.
Indeed, this aspect was not corrected, obviously we have made a mistake in its correction. Now it is corrected. Section 4. Discussion (ln.274) and section 5 Conclusions (ln. 374).